# EAPB0503, an Imidazoquinoxaline Derivative Modulates SENP3/ARF Mediated SUMOylation, and Induces NPM1c Degradation in NPM1 Mutant AML

**DOI:** 10.3390/ijms23073421

**Published:** 2022-03-22

**Authors:** Hala Skayneh, Batoul Jishi, Rita Hleihel, Maguy Hamie, Rana El Hajj, Carine Deleuze-Masquefa, Pierre-Antoine Bonnet, Marwan El Sabban, Hiba El Hajj

**Affiliations:** 1Department of Experimental Pathology, Immunology and Microbiology, Faculty of Medicine, American University of Beirut, P.O. Box 11-0236, Riad El-Solh, Beirut 1107 2020, Lebanon; has82@mail.aub.edu (H.S.); mh242@aub.edu.lb (M.H.); 2Department of Anatomy, Cell Biology and Physiological Sciences, Faculty of Medicine, American University of Beirut, P.O. Box 11-0236, Riad El-Solh, Beirut 1107 2020, Lebanon; bhj02@mail.aub.edu; 3Department of Internal Medicine, Faculty of Medicine, American University of Beirut, P.O. Box 11-0236, Riad El-Solh, Beirut 1107 2020, Lebanon; rh150@aub.edu.lb; 4Department of Biological Sciences, Beirut Arab University, P.O. Box 11-0236, Riad El-Solh, Beirut 1107 2020, Lebanon; r.hajj@bau.edu.lb; 5CNRS, Institute of Biomolecules Max Mousseron IBMM, Université de Montpellier Unité de Recherche UMR5247, École Nationale Supérieure de Chimie de Montpellier, Faculty of Pharmacy, Montpellier University, 34090 Montpellier, France; carine.masquefa@umontpellier.fr (C.D.-M.); pierre-antoine.bonnet1@umontpellier.fr (P.-A.B.)

**Keywords:** NPM1c AML, immunomodulatory drugs, post-translational modifications, ARF/NPM1/SENP3, *P*53/HDM2

## Abstract

Nucleophosmin-1 (NPM1) is a pleiotropic protein involved in numerous cellular processes. NPM1 shuttles between the nucleus and the cytoplasm, but exhibits a predominant nucleolar localization, where its fate and functions are exquisitely controlled by dynamic post-translational modifications (PTM). Sentrin/SUMO Specific Peptidase 3 (SENP3) and ARF are two nucleolar proteins involved in NPM1 PTMs. SENP3 antagonizes ARF-mediated NPM1 SUMOylation, to promote ribosomal biogenesis. In Acute Myeloid Leukemia (AML), NPM1 is frequently mutated, and exhibits an aberrant cytoplasmic localization (NPM1c). NPM1c mutations define a separate AML entity with good prognosis in some AML patients, rendering NPM1c as a potential therapeutic target. SENP3-mediated NPM1 de-SUMOylation induces resistance to therapy in NPM1c AML. Here, we demonstrate that the imidazoquinoxaline EAPB0503 prolongs the survival and results in selective reduction in the leukemia burden of NPM1c AML xenograft mice. Indeed, EAPB0503 selectively downregulates HDM2 expression and activates the *p*53 pathway in NPM1c expressing cells, resulting in apoptosis. Importantly, we unraveled that NPM1c expressing cells exhibit low basal levels of SUMOylation paralleled with high SENP3 and low ARF basal levels. EAPB0503 reverted these molecular players by inducing NPM1c SUMOylation and ubiquitylation, leading to its proteasomal degradation. EAPB0503-induced NPM1c SUMOylation is concurrent with SENP3 downregulation and ARF upregulation in NPM1c expressing cells. Collectively, these results provide a strong rationale for testing therapies modulating NPM1c post-translational modifications in the management of NPM1c AML.

## 1. Introduction

*Nucleophosmin-1 (NPM1)* encodes for a chaperone phosphoprotein [1], which shuttles between the nucleus and cytoplasm, but mostly resides in the nucleolar compartment [2,3]. NPM1 is involved in a host of cellular processes. These include ribosomal biogenesis, 2-O-methylation of rRNA, histone chaperoning, “liquid–liquid” phase separation of the nucleolus, centrosome duplication control, DNA repair, *P*53 activation in response to stress stimuli and p14^Arf^ (ARF) stabilization [1,4,5,6,7,8,9,10].

NPM1 is dynamically modulated by reversible post-translational modifications, among which SUMOylation and de-SUMOylation, dictate its fate and function. Indeed, ARF binds NPM1 and antagonizes NPM1-mediated ribosomal biogenesis by inducing its SUMOylation, ubiquitination and proteosomal degradation [11]. Conversely, NPM1 de-SUMOylation promotes rRNA synthesis and is catalyzed by the nucleolar SUMO-specific protease SENP3 [12,13].

*NPM1* mutations account for around one third of patients with Acute Myeloid Leukemia (AML) [1,14]. AML is a genetically heterogeneous and complex blood malignancy accounting for around 80% of acute leukemias in adults [15]. AML is characterized by a clonal expansion of myeloid precursors resulting in the aberrant accumulation of myeloblasts in the bone marrow (BM) [16]. Elderly patients have a poor prognosis, where most patients succumb within 2 years [17]. More than 20 driver recurrent mutations were unveiled in AML patients, the most frequent of which occur in *Nucleophosmin 1* (*NPM1*) and *Fms*-*Like Tyrosine Kinase 3 (FLT3)* [18,19,20]. *NPM1* mutations lead to the creation of a de novo nuclear export signal [1,14,21] resulting in the aberrant and continuous cytoplasmic export of NPM1c, contributing to AML leukemogenesis [14]. NPM1c impact both the prognosis and the response to treatment in AML [19,22].

The standard induction treatment of AML relies on the combination of Cytarabine with Daunorubicin or Idarubicin (“7 + 3” regimen) [23]. Complete remission is achieved in almost 70% of treated patients; however, relapse represents the major cause of treatment failure [24]. Currently, the clinical management of AML started shifting towards incorporating targeted therapies against key driving mutations [10,25,26,27,28,29,30,31]. Many of these targeted therapies are still under pre-clinical or clinical investigation [32,33,34]. *NPM1* mutations, in absence of or with low allelic frequency ratio FLT3-internal tandem duplication (ITD), convey a relatively favorable prognosis [18,20,31,35]. Moreover, the post-translational modifications of NPM1 affect the therapeutic response in AML, whereby SENP3-mediated deSUMOylation of NPM1 induced the resistance of AML cells to chemo- and radiotherapy [36]. We and others demonstrated that drugs targeting NPM1c induce a selective growth arrest and apoptosis in *NPM1c* AML cells [25,26,37,38,39]. Particularly, we demonstrated that the imidazoquinoxaline EAPB0503, an Imiquimod analog, induced the selective growth arrest and apoptosis in *NPM1c* AML cell lines and in ex vivo treated blasts from *NPM1c* AML patients. This was subsequent to NPM-1c proteasomal degradation, and the restoration of the *wt*-NPM-1 nucleolar localization in vitro. Notably, EAPB0503 selectively reduced leukemia burden in *NPM1c* AML xenograft mice [39]. In this study, we dissected the molecular mechanisms of EAPB0503 efficacy, and demonstrated that this drug prolongs the survival of *NPM1c* AML xenograft mice. We also unraveled that NPM1c expressing cells exhibit low basal levels of SUMOylation and ubiquitylation. The low SUMO levels were accompanied by high SENP3 and low ARF basal levels. Mechanistically, EAPB0503 selectively activated the *P*53 pathway in the NPM1c expressing OCI-AML3 cell line and in ex vivo treated blasts from *NPM1c* AML patients. Strikingly, EAPB0503 induced NPM1c SUMOylation and subsequent ubiquitylation, leading to its proteasomal degradation. NPM1c SUMOylation was concurrent with early SENP3 downregulation and ARF upregulation in NPM1c expressing cells. Collectively, these findings further strengthen the idea of targeting NPM-1c to eradicate leukemic blasts and highlight the importance of reverting SENP3-mediated de-SUMOylation of NPM1c to induce its degradation. Altogether, our results warrant a broader preclinical then clinical evaluation of EAPB0503 as targeted therapeutic strategy in *NPM1c* AML.

## 2. Results

### 2.1. EAPB0503 Prolongs the Survival of Xenograft Mice with NPM1c-Expressing Cells, Attenuating the Pathophysiological Features of NPM1c AML In Vivo

We have previously demonstrated the selective in vivo efficacy of EAPB0503 on *NPM1c* AML leukemia burden in a xenograft mouse model [39]. Here, we examined the effect of EAPB0503 on the survival and organ infiltration in *NPM-1c* AML xenograft mice. Eight-week-old NSG mice were intravenously injected with OCI-AML2 or OCI-AML3 cells. Seven days post injection, xenograft mice were treated intraperitoneally with EAPB0503 every other day over a period of 3 weeks (Figure 1A). While untreated control mice, or OCI-AML2 treated mice succumbed at day 40, EAPB0503 selectively and significantly prolonged the survival in OCI-AML3 xenograft mice for up to 100 days (*p* value = 0.003) (Figure 1B).

AML patients are at higher propensity to suffer from liver failure [40]. We examined the liver gross pathology and histopathology following treatment with EAPB0503. While gross pathology of the liver revealed pale color and white nodules in untreated xenograft mice, EAPB0503 treatment showed a normal gross macroscopy selectively in livers of OCI-AML3 xenograft mice (Figure 1C). Consistent with these results, H&E stain showed a clear infiltration of the liver in untreated xenograft mice injected with OCI-AML3, while EAPB0503 treatment preserved the normal architecture of the liver, with a very low number of infiltrating leukemic cells (Figure 1D).

To understand the molecular basis of this prolonged survival, BM cells of femurs and tibias were flushed and stained for hCD45, a prototypic receptor-like protein tyrosine phosphatase expressed on all nucleated hematopoietic cells [41,42]. As previously reported [39], OCI-AML3 burden in the BM of xenograft mice was significantly reduced from 47% to 25% upon EAPB0503-treatment (*p* < 0.05) (Figure 1E), as compared to OCI-AML2 burden (24% in untreated versus 34% in EAPB0503 treated mice) (Figure 1E).

Given that EAPB0503 prolonged the survival and reduced the BM leukemic burden exclusively in OCI-AML3 xenograft mice, we tested the expression protein level of NPM1c in the BM of treated versus untreated mice. Consistent with the beneficial selective effect of EAPB0503 against *NPM1c* AML xenograft mice, NPM1c expression was abolished in the BM of OCI-AML3 xenograft mice (Figure 1F). Altogether, these results illustrate the potency of EAPB0503 on *NPM1c* AML in vivo.

### 2.2. EAPB0503 Activates the p53 Signaling Pathway and HDM2 Downregulation in NPM-1c AML Cells

We have previously demonstrated that EAPB0503 induces apoptosis exclusively in the *NPM-1c* OCI-AML3 cell line, 48 h after in vitro treatment, following a substantial upregulation of total *p*53 protein levels, and its phosphorylated form *P*-*p*53 [39]. We expanded our results to primary blasts from patients with *wt-NPM1* or *NPM1c* AML. Since the human double min 2 protein (HDM2) is a ubiquitin E3 ligase and a major endogenous negative regulator of *p*53, which leads to its proteasomal degradation [43], we investigated the effect of EAPB0503 on HDM2. First, our results indicate that HDM2 basal protein levels are higher in OCI-AML3 expressing NPM1c, as compared to OCI-AML2 expressing *wt*-NPM1 (Figure 2A). These observed higher HDM2 levels were consistent with lower *p*53 protein levels and unphosphorylated *p*53 in OCI-AML3 (Figure 2A). Treatment of OCI-AML3 with EAPB0503 revealed a gradual decrease of HDM2 protein levels reaching significance at both 24 h and 48 h post treatment (*p* value = 0.0216 and 0.0002 respectively). The maximum downregulation was obtained selectively in OCI-AML3 cell line, at 48 h post treatment (*p* < 0.001). This decrease was paralleled with a sharp and significant activation of the *p*53 pathway, as demonstrated by the ratio *P*-*p*53/p53 at the same time points (*p* value = 0.0001 and 0.0002 respectively). Notably, the activation of the *p*53 pathway reached its maximum at 24 h post treatment, exclusively in OCI-AML3 cells with EAPB0503 (*p* < 0.001), with no significant effect in OCI-AML2 cell line (Figure 2B). Interestingly, basal levels of HDM2, *p*53 and *P*-*p*53 in primary blast from 3 out of 4 *NPM1c* AML patients exhibited a similar protein expression profile as that seen in OCI-AML3 (Figure 2C). Ex vivo treatment of these primary blasts with EAPB0503 led to an early activation of the *p*53 pathway, as compared to OCI-AML3. Indeed, 6 h post treatment with EAPB0503, a significant decrease in the expression of HDM2 protein (*p* value = 0.0065) accompanied by a significant activation of the *p*53 pathway (*p* value = 0.021) were observed (Figure 2D). A sharp and significant activation of this pathway was sustained after 24 h post treatment with EAPB0503, selectively in primary blasts from *NPM1c* AML patients (Figure 2E). Collectively, these results demonstrate that NPM1c inhibits *P*-*p*53 concomitant to an upregulation of the *p*53 ubiquitin ligase HDM2, presumably inhibiting apoptosis and conferring survival properties to these cells. Targeting NPM1c by EAPB0503 activates the *P*53 pathway and downregulates HDM2, hence orchestrating the pro-apoptotic activity and inducing apoptosis in EAPB0503 treated *NPM1c* AML cells.

### 2.3. NPM1c Expressing Cells Exhibit Low Basal Levels of SUMOylation, and EAPB0503 Restores NPM1c Post-Translational Modifications Triggering its Proteasomal Degradation

We previously demonstrated that EAPB0503 induces the proteasomal degradation of NPM1c and the nucleolar re-localization of *wt*-NPM1 48 h post treatment [39]. The early activation of the *p*53 pathway prompted us to investigate the effect of EAPB0503 on NPM1c expression at earlier time points. Interestingly, EAPB0503 triggered NPM1c downregulation in a time dependent manner and NPM1c degradation was initiated as early as 6 h post treatment in OCI-AML3 cells (*p* < 0.001) (Figure 3A).

We then examined the effect of EAPB0503 on NPM1 post-translational modifications, namely SUMOylation and ubiquitylation. We used the proximity ligation Duolink assays (PLA) to assess the endogenous SUMO2/3 conjugation with NPM1 in OCI-AML3 and OCI-AML2 cells. Remarkably, in OCI-AML3 cells, conjugation between SUMO2/3 and NPM1 was not detected in untreated cells, while this conjugation was observed in OCI-AML2. Importantly, 6 h of treatment with EAPB0503 restored SUMO2/3-NPM1 conjugation in OCI-AML3 cells (Figure 3B). Similar results were obtained using the immunoprecipitation assay. Indeed, we could not detect basal SUMOylation levels of NPM1 in untreated OCI-AML3 (Figure 3C), demonstrating that SUMOylation levels are altered in NPM1c expressing cells. Furthermore, 6 h of treatment with PS-341 partially restored NPM1 SUMOylation while 6 h of treatment with EAPB0503 alone or combined with the proteasome inhibitor PS-341 substantially increased NPM1 SUMOylation in these cells (Figure 3C). Similar to SUMOylation, basal levels of NPM1 ubiquitylation were not detected in OCI-AML3 (Figure 3D), underpinning an effect of these two post-translational modifications on the leukemogenesis of *NPM1c* AML. Importantly, 24 h post treatment with EAPB0503 markedly restored NPM1 ubiquitylation and this effect was further enhanced following treatment with PS-341 alone or combined with EAPB0503 (Figure 3D). Altogether, our results indicate that NPM1c expressing cells exhibit altered SUMOylation and ubiquitylation profiles, potentially contributing to survival of leukemic blasts. Moreover, EAPB0503 could revert these post-translational modifications indicating that EAPB0503-induced NPM1c downregulation is secondary to oncoprotein degradation.

### 2.4. EAPB0503 Reduces SENP3 Levels, Increases ARF Levels and Restores NPM1 SUMOylation in NPM1c AML

Under physiological conditions, NPM1 is involved in ribosomal biogenesis through a balance between SENP3 and ARF [12]. Indeed, an interplay between NPM1, SENP3 and ARF happens via cycles of SUMOylation/deSUMOylation to induce ribosomal biogenesis. It is well documented that ARF SUMOylates NPM1 prohibiting ribosomal biogenesis, while SENP3 deSUMOylates NPM1 switching ribosomal biogenesis on [12,44]. In *NPM1c* AML cells, basal ARF levels are very low [45,46], due to NPM1c failure to stabilize and retain ARF in the nucleolus. We first screened the effect of NPM1c on the basal levels of SENP3 and ARF in NPM1c cell lines and in primary blasts from *NPM1c* AML patients. Strikingly, SENP3 protein levels were particularly upregulated in all screened NPM1c, but *wt*-NPM1 expressing cells (Figure 4A,B), presumably explaining the low basal SUMOylation levels in these cells (Figure 3C,D). Remarkedly, treatment with EAPB0503 induced an antagonistic effect on SENP3 and ARF proteins. Indeed, a gradual and significant decrease of SENP3 expression was obtained following 24 h and reached less than 20% after 48 h of treatment of OCI-AML3 with EAPB0503 (*p* values = 0.033 and 0.001 respectively) (Figure 4C). This decrease was accompanied with an upregulation of ARF protein levels at as early as 6 h (Figure 4C). Similar results were obtained on primary blasts from three NPM1c AML patients following ex vivo treatment with EAPB0503 as compared to primary blasts from *wt*-NPM1 AML patients. Indeed, a significant decrease in SENP3 levels (*p* value = 0.01) concomitant with a sharp and significant increase in ARF protein levels was obtained as early as 6 h post treatment with EAPB0503 (Figure 4D), and this effect was sustained at 24 h post treatment with this drug (*p* value = 0.0003) (Figure 4E). Collectively, these results presumably indicate that NPM1c upregulates SENP3 and downregulates ARF, to sustain an NPM1 de-SUMOylation profile, which induces ribosomal biogenesis and contributes to leukemogenesis. Moreover, treatment with EAP0503 induced SUMOylation of NPM1c, following upregulation of ARF, and downregulation of SENP3, presumably to inhibit ribosomal biogenesis in these cells.

## 3. Discussion

*NPM1* mutations are encountered in one third of AML patients [1,14] and represent one of the most frequent mutations [18,19,20]. *NPM1* mutations impact AML prognosis and its therapeutic response [19,22]. In patients with normal karyotype, NPM1 when present alone confers a better prognosis [47]. Indeed, *NPM1c* AML cells proved addicted to the continuous expression of NPM1c for their survival [37], and NPM1c is a potential target in this category of AML patients. Notably, targeted therapies triggering NPM1c degradation yielded promising preclinical and clinical outcomes. In that sense, the addition of retinoic acid and arsenic trioxide synergistically mediated the proteasomal degradation of NPM1c in AML cell lines and in primary blasts from *NPM1c* AML patients, leading to differentiation and apoptosis [25,26]. More importantly, retinoic acid and arsenic treatment significantly reduced blasts in some *NPM1c* AML patients who were unfit to chemotherapy [25,38]. Likewise, Actinomycin D induced complete remissions in *NPM1c* AMLs [28,48,49], and this clinical efficacy happened via targeting mitochondria, boosting reactive oxygen species production, hence restoring senescence of NPM1c expressing cells. Dual targeting of mitochondria with actinomycin D and the BCL-2 inhibitor venetoclax sharply potentiated the in vivo anti-AML activities of these drugs [27]. Venetoclax alone, or combined to low intensity chemotherapy, Arsenic or menin inhibitors proved efficient in preclinical and more importantly in different categories of *NPM1c* AML patients including those with minimal residual disease, or relapsed refractory patients [29,30,50,51,52,53]. BET-inhibitors also induced differentiation and apoptosis in NPM1c cells following induced proteasome-dependent degradation of NPM1c [54]. In line with these findings, we demonstrated that EAPB0503, induced the selective growth arrest and apoptosis in *NPM1c* AML cells following NPM1c proteasomal degradation [39]. In this study, we deciphered the molecular mechanisms associated with NPM1c degradation and its subsequent induced cancer cell death. We showed that this beneficial efficacy is accompanied by the degradation of NPM1c in vivo, yielding an improvement of organ gross pathology and histology and more importantly, inducing the selective prolonged survival of EAPB0503-treated xenograft *NPM1c* AML mice. We also showed that EAPB0503 leads to a progressive degradation of NPM1c as early as 6 h post treatment. This seems to be involved in triggering early mechanisms, ultimately leading to cell death of NPM1c-expressing cells. We also broadened our findings on *p*53 activation [39], to characterize the time point and the other important players in the *p*53 pathway in EAPB0503-treated cell lines and ex vivo treated blasts. While NPM1c degradation started at 6 h, *p*53 activation through HDM2 degradation was not observed before 24 h post treatment in the OCI-AML3 cell line. Interestingly, in freshly isolated primary blasts from NPM1c AML patients, an upregulation of basal levels of HDM2 concomitant with a downregulation of *p*53 was obtained. Treatment with EAPB0503 reverted the protein expression profiles as early as 6 h, and ultimately inducing apoptosis at later time points. Our results demonstrate the importance of the *p*53/HDM2 pathway in the leukemogenesis of NPM1c AML and its response to targeted therapies. This finding is in line with several studies where therapeutics exert their anti-cancer activity in general and in AML in particular, through restoring *p*53 activity and downregulating HDM2 expression [55,56,57,58,59,60,61].

SUMOylation and ubiquitylation are dynamic and reversible modifications determining the function and fate of proteins and mediating key cellular processes. These modifications are tightly involved in pathophysiological conditions. For instance, the deregulation of SUMOylation was reported in several cancers and triggers cancer cells growth under stress conditions [62]. Moreover, if the SUMOylation process of proteins is affected by variations in SENP3 levels, abnormal cellular activities will be encountered [63]. In line with these findings, we demonstrated that *NPM1c* AML cell lines and primary AML blasts from *NPM1c* AML patients exhibit low basal levels of SUMOylation, presumably playing a role in their growth and proliferation. We also demonstrated that *NPM1c* AML cell lines and primary AML blasts from *NPM1c* AML patients express upregulated basal levels of SENP3, concomitant with low basal levels of ARF. Indeed, NPM1 SUMOylation and de-SUMOylation regulate ribosomal biogenesis through a tight balance of interaction with either ARF or SENP3 which antagonize each other [12]. While ARF-mediated NPM1 SUMOylation inhibits ribosomal biogenesis, SENP3-mediated NPM1 de-SUMOylation promotes ribosomal biogenesis [11,12,13,64]. These findings presumably delineate that, in the context of *NPM1c* AML, NPM1c de-SUMOylation is likely sustained via upregulation of SENP3, hence counteracting ARF-mediated NPM1c SUMOylation, to promote ribosomal biosynthesis, protein synthesis and leukemic blast survival.

Triggering oncoproteins post-translational modifications to induce their proteasomal degradation represents a major approach in the molecular efficacy of targeted therapies [65]. In AML, the post-translational modifications of NPM1 affect the therapeutic response and SENP3-mediated deSUMOylation of NPM1 induces the resistance of AML cells to chemo- and radiotherapy [36]. In agreement with this study, our results reveal that EAPB0503 is downregulating SENP3, as part of its preclinical efficacy. Indeed, EAPB0503-induced deregulation of SENP3 is accompanied by ARF upregulating, hence the restoration of SUMO2/3 conjugated NPM1c in NPM1c AML cells. This event was followed by NPM1c ubiquitylation and degradation. EAPB0503 degrades NPM1c oncoprotein, in a proteasome-dependent manner and in accordance with several studies demonstrating that targeting NPM1c oncoprotein proteasomal degradation inhibits the proliferation and induces cell death of NPM1c AML leukemic cells [25,26,37,39].

In conclusion, our results unravel aberrant post-translational modification in NPM1c AML. Furthermore, we dissected the molecular mechanisms of EAPB0503 efficacy and demonstrated that targeting NPM1c post-translational modifications to trigger its degradation holds promising therapeutic expectations in *NPM1c* AML.

## 4. Materials and Methods

### 4.1. Cell Lines and Viability

OCI-AML2 (expressing *wt*-NPM1, from Dr. H. de Thé) and OCI-AML3 cells (expressing NPM1c, from Dr. D. Bouscary) were grown in minimum essential medium alpha (MEM-α) supplemented with 20% fetal bovine serum. Primary AML cells from patients’ BM were extracted as described by [25] after approval by the Institutional Review Board at the American University of Beirut and after consented agreement of patients according to Helsinki’s Declaration. Cells were seeded at a concentration of 2 × 10^5^/mL. Cell growth was assessed using the trypan blue exclusion dye assay. EAPB0503 [66,67,68] was dissolved in dimethylsulfoxide (DMSO) (Amresco, Solon, OH, USA) at a stock solution of at 0.1 M, aliquoted, and stored at −20 °C. EAPB0503 was used at a concentration of 1 μM as described [39]. The proteasome inhibitor (PS-341) was used at a concentration of 10 nM. Cell viability was assessed at three different time points of treatment (6, 24 and 48 h).

### 4.2. Immunoblotting

After 6, 24 or 48 h of treatment with EAPB0503, protein extracts from in vitro treated cell lines or ex vivo treated primary blasts from AML patients were probed with the following antibodies: NPM1c (PA1-46356, 1:1000, Thermo Fisher Scientific, Waltham, MA, USA), ARF (MA5-14,260, 1:200, Thermo Fisher Scientific, Waltham, MA, USA), SENP3 (5591, 1:1000, Cell Signaling, Danvers, MA, USA), *P*-*p*53 (Ser15) (9284, 1:1000, Cell Signaling, Danvers, MA, USA), HDM2 (ab16,895, 1:250, Abcam, Cambridge, UK), NPM1 (*wt* + c) (ab10,530, 1:1000, Abcam, Cambridge, UK), *p*53 (sc-126, 1:200, Santa Cruz Biotechnology, Dallas, TX, USA), SUMO2/3 (sc-32,873, 1:200, Santa Cruz Biotechnology, Dallas, TX, USA), and Ubiquitin (sc-166,553, dilution, Santa Cruz Biotechnology, Dallas, TX, USA) before incubation with the monoclonal horseradish peroxidase (HRP)-conjugated secondary antibodies. Loading control was performed by probing with the mouse HRP-conjugated Glyceraldehyde 3-phosphate dehydrogenase (GAPDH) antibody (MAB5476, 1:5000, Abnova, Taipei, Taiwan), beta-actin Actin (8H10D10,1:1000, Cell Signaling, Danvers, MA, USA), or Histone (H3) (ab1791, 1:2000, Abcam, Cambridge, UK). Immunoblots were detected using the luminol detection kit (Santa Cruz) and images were captured using the XOMAT or BioRad Chemidoc MP system.

### 4.3. Immunoprecipitation

OCI-AML2 and OCI-AML3 cell pellets were washed in ice-cold phosphate-buffered saline supplemented with 10 mM *N*-ethylmaleimide prior to lysis in 2% sodium dodecyl sulfate and 50 mM Tris, pH 8. After brief sonication, cell lysates were diluted 10-fold in immuno-precipitation (IP) buffer containing 50 mM Tris, pH 8, 200 mM NaCl, 0.1 mM EDTA, 0.5% NP-40, 10% glycerol, and protease inhibitors. Lysates were incubated overnight with NPM1 (*wt* + c) antibody. Beads were washed 3 times in the IP buffer prior to elution of immuno-precipitated proteins in sample buffer. Cell extracts were separated on 4% to 12% gradient gels (Invitrogen, Waltham, MA, USA) as described [38].

### 4.4. In Situ Proximity Ligation Assays (Duolink), and Confocal Microscopy

OCI-AML2 and OCI-AML3 were washed with PBS and fixed with ice-cold methanol at −20 °C overnight, then cytospun onto glass slides. Protein–protein interactions were visualized using the Duolink in situ proximity ligation assay (PLA) system (Olink Bioscience, Uppsala, Sweden) as described [69]. Assays were performed using anti-NPM1 (*wt* + c) and SUMO2/3 primary antibodies following the manufacturer’s instructions. Staining of nuclei was performed with 4′, 6-diamidino-2-phenylindole (DAPI) (Invitrogen, Waltham, MA, USA). Images were acquired by confocal microscopy using a Zeiss LSM710 confocal microscope (Zeiss, Oberkochen, Germany), and images were processed using Zen 2009 (Carl Zeiss, Oberkochen, Germany).

### 4.5. Xenograft Animal Studies

NOD/Shi-*scid IL2r-gamma*^−/−^ mice (NSG) mice were obtained from Jackson Laboratories (USA). All mouse protocols were approved by the Institutional Animal Care and Utilization Committee of the American University of Beirut. 3 × 10^6^ OCI-AML3 or OCI-AML2 cells were injected into the tail vein of eight-week-old mice (12 mice per group). After 7 days post AML cells’ injection, mice were treated intraperitoneally with EAPB0503 (2.5 mg/kg, 50 μg/mouse) every other day over a period of 3 weeks. EAPB0503 was dissolved in DMSO and diluted in equal volume of lipofundin (vehicle) before intraperitoneal administration to mice [70,71]. Six mice per condition were sacrificed for bone marrow flushing, CD45 staining, western blot analysis and organ infiltration, while the remaining six mice were kept to monitor survival.

To monitor leukemia burden in mice, human CD45 staining was performed. Briefly, three weeks post treatment with EAPB0503, BM cells were flushed from the femurs and tibias of euthanized animals. Cell surface staining was performed on 100 μL of the sample using 20 μL of anti-human CD45 PerC-P antibody (BD#345809). After incubation for 15 min in the dark, erythrocytes were lysed using 1 mL FACS Lyse (BD). Labeled samples were washed twice and analyzed on a Guava flow cytometer.

To assess organ infiltration, livers from either treated or untreated mice were fixed in neutral buffer formalin (Sigma-Aldrich, Darmstadt, Germany), embedded in paraffin, sectioned, stained with hematoxylin and eosin, and histopathology was examined by light microscopy.

## Figures and Tables

**Figure 1 ijms-23-03421-f001:**
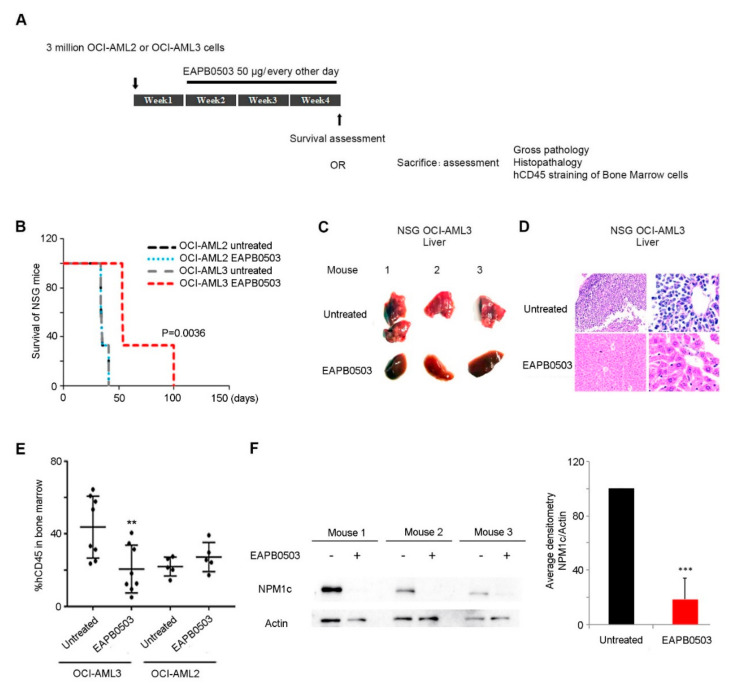
EAPB0503 prolongs survival and attenuates AML pathological features in NPM1c AML xenograft NSG mice. (**A**) Eight-week-old NSG mice were injected with 3 million OCI-AML2 or OCI-AML3 cells intravenously (12 mice per cell line per condition). After 1 week post injection, EAPB0503 (2.5 mg/kg, 50 µg/mouse) was intraperitoneally administered every other day, for a total period of 3 weeks. At the end of week 3, one group of six mice per condition was monitored for survival. The remaining six mice per condition were sacrificed for gross pathology, histopathology and hCD45 staining on the flushed bone marrow cells. (**B**) Kaplan–Meier overall survival of untreated NSG mice injected with OCI-AML2 or OCI-AML3 (n = 6, black line and gray line, respectively) or treated with EAPB0503 (n = 6, blue line and red line, respectively). The *t*-test was performed to validate significance. *P*-values less than 0.05 were considered significant. (**C**) Gross pathology of livers from three representative untreated (upper panel) or EAPB0503-treated (lower panel) OCI-AML3 xenograft mice. (**D**) Histological analysis (H&E stain) of the liver of a representative untreated (upper panel) or EAPB0503 treated (lower panel) OCI-AML3 xenograft mouse (left panel, magnification 10×, right panel, magnification 40×). (**E**) Graph showing the hCD45 PerCP percentage in untreated or in EAPB0503 treated xenograft mice injected with OCI-AML3 or OCI-AML2 (n = 6 per condition). (**F**) Western blot of NPM1c in BM cells extracted from NSG mice xenografted with OCI-AML3 (three representative mice out of six are shown), after in vivo treatment with EAPB0503. Histogram shows the average densitometry of NPM1c/Actin of the 3 representative mice. The *t*-test was performed to validate significance. ** and *** indicate *p* values ≤ 0.05; 0.01 and 0.001, respectively. *p*-values less than 0.05 were considered significant.

**Figure 2 ijms-23-03421-f002:**
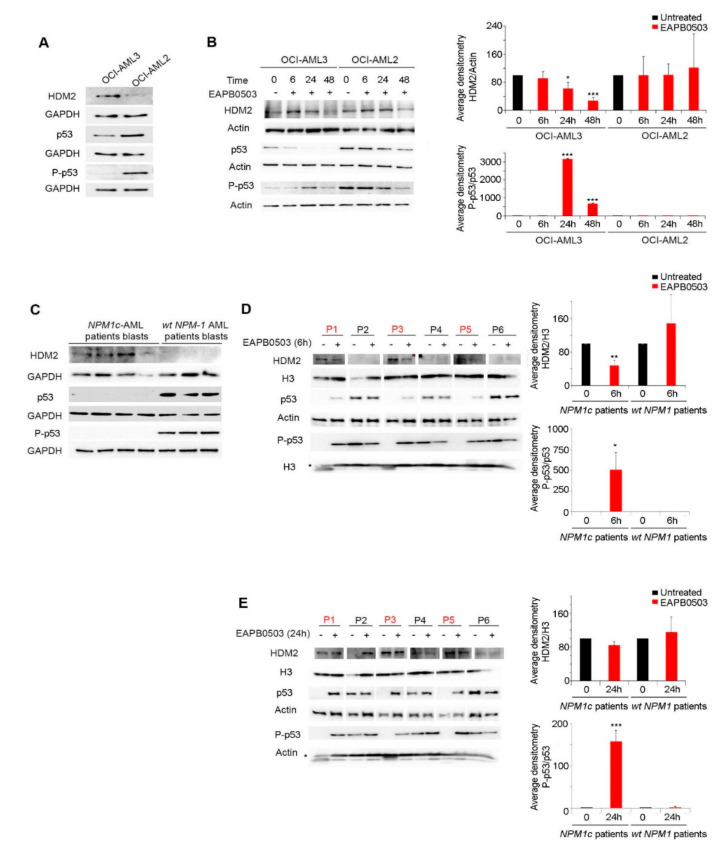
EAPB0503 actives *p*53 signaling pathway and dowregulates its ubiquitin ligase HDM2. (**A**) Western blot analysis showing basal levels of HDM2, *p*53, *P*-*p*53, GAPDH and actin in untreated OCI-AML3 and OCI-AML2 cells. (**B**) Western blot analysis of HDM2, *p*53, *P*-*p*53, and actin in EAPB0503 treated OCI-AML3 and OCI-AML2 cells for 6, 24 and 48 h. Histograms represent the densitometries of HDM2/Actin and *P*-*p*53/*p*53 ratio of 3 independent experiments. (**C**) Western blot analysis showing the basal expression levels of HDM2, *p*53 and *P*-*p*53 with respect to GAPDH in blasts from NPM1c and *wt*-NPM1 AML patients. Western blot analysis shows the expression levels of HDM2, and *P*-*p*53 with respect to total histone H3, and *p*53 with respect to Actin in blasts from NPM1c (red) and *wt*-NPM1 (black) AML patients following ex vivo treatment with EAPB0503 for 6 h (**D**) or (**E**) 24 h. Histograms represent the average densitometries of HDM2/H3 and *P*-*p*53/*p*53 ratio of the three NPM1c and three *wt*-NPM1 patients represented in the Western blots.*, ** and *** indicate *p* values ≤ 0.05; 0.01 and 0.001, respectively. *p*-values less than 0.05 were considered significant.

**Figure 3 ijms-23-03421-f003:**
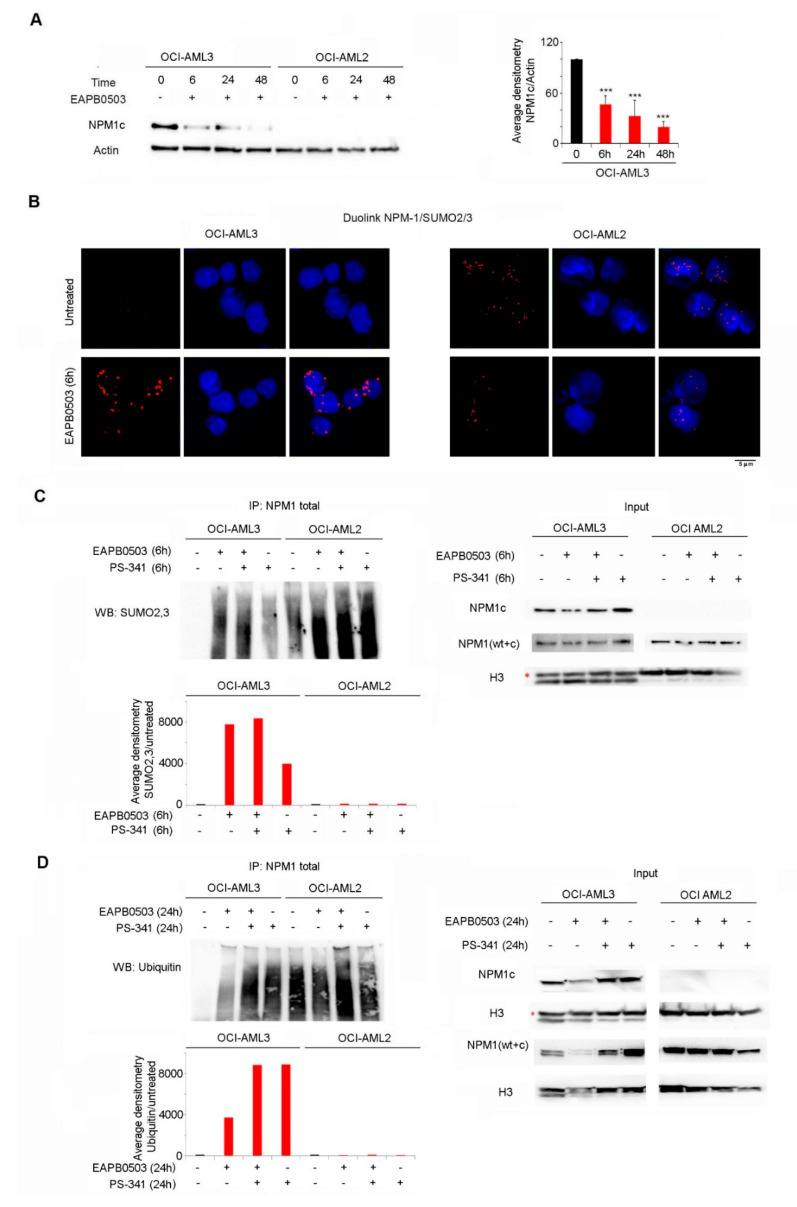
NPM1c expressing cells exhibit low basal levels of SUMOylation, and EAPB0503 restores NPM1c post-translational modifications triggering its proteasomal degradation. (**A**) Western blot analysis of NPM1c and Actin in OCI-AML3 and OCI-AML2 cells treated with EAPB0503 for 6, 24 and 48 h. Histogram represents the average densitometry of NPM1c/Actin in OCI-AML3 cells in three independent experiments. (**B**) Endogenous interactions detected by Duolink between NPM1 (*wt* + c) and SUMO 2/3 (Red) in untreated OCI-AML3 and OCI-AML2 cells (upper panel) or EAPB0503 treated cells for 6 h (lower panel). Nuclei were stained with 4-,6-diamidino-2-phenylindole (DAPI) (blue). (**C**) OCI-AML3 and OCI-AML2 cells were treated with EAPB0503 and PS-341 for 6 h. NPM1 (*wt* + c) immunoprecipitates [IP-NPM1 (*wt* + c)] were blotted for SUMO 2/3. Input was blotted for NPM1c, NPM1 (*wt* + c) and H3. (**D**) OCI-AML3 and OCI-AML2 cells were treated with EAPB0503 and PS-341 for 24 h. NPM1 (*wt* + c) immunoprecipitates [IP-NPM1 (*wt* + c)] were blotted for Ubiquitin. Input was blotted for NPM1c, NPM1 (*wt* + c) and H3. *** *p* < 0.001.

**Figure 4 ijms-23-03421-f004:**
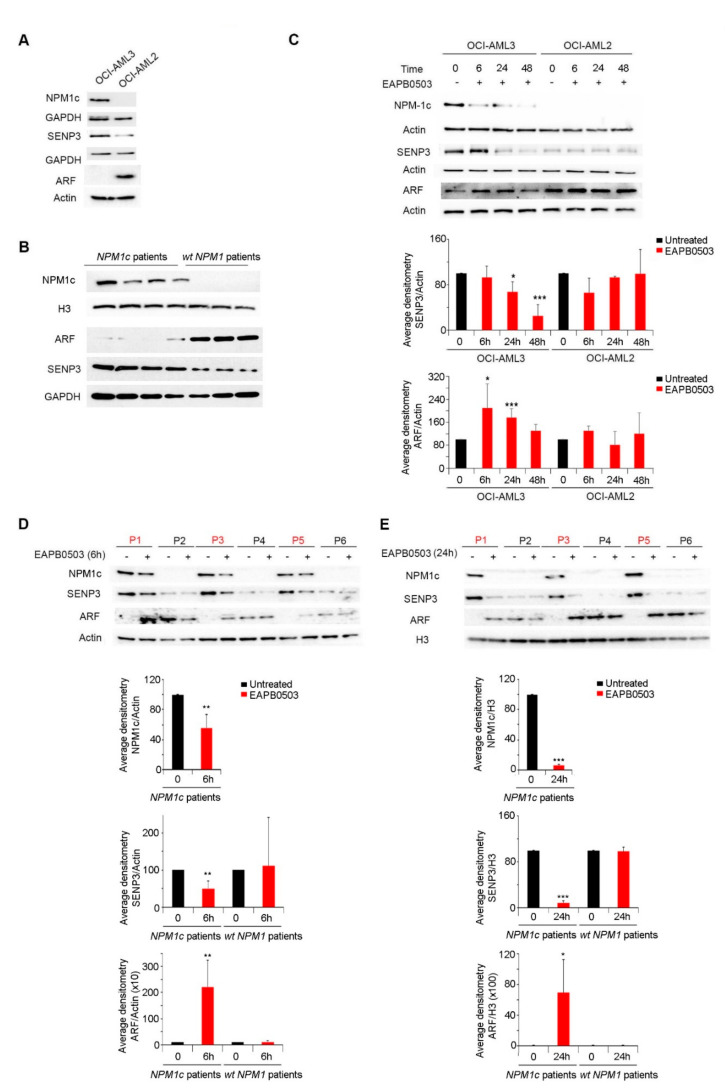
EAPB0503 effect on NPM1c post-translational modification is mediated through the SENP3 degradation and *p*14^ARF^ upregulation. Western blot analysis of basal levels of NPM1c, SENP3, ARF in untreated OCI-AML3 and OCI-AML2 cells (**A**) and in blasts from *NPM1c* or *wt-NPM1* AML patients (**B**). Western blot analysis of NPM1c, SENP3, ARF in EAPB0503 treated OCI-AML3 and OCI-AML2 cells for 6, 24 and 48 h (**C**). Histograms represent the densitometries of SENP3/Actin and ARF/H3 in 3 independent experiments. Western blot analysis of NPM1c, SENP3, ARF in blasts from patients with NPM1c (red) or *wt*-NPM1 (black) AML patients following ex vivo treatemnt with EAPB0503 for 6 h (**D**) or 24 h (**E**). Histograms represent the average densitometries of NPM1c/actin, SENP3/actin and ARF/H3 of the three NPM1c and three *wt*-NPM1 patients represented in the Western blots. *, ** and *** indicate *p* values ≤ 0.05; 0.01 and 0.001, respectively. *P*-values less than 0.05 were considered significant.

## Data Availability

Not applicable.

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
