# Peer review of "EAPB0503, an Imidazoquinoxaline Derivative Modulates SENP3/ARF Mediated SUMOylation, and Induces NPM1c Degradation in NPM1 Mutant AML"

_ijms, 2022, doi:10.3390/ijms23073421_

Round 1
Reviewer 1 Report
Overall, this is a nice paper that attempts to elucidate the molecular mechanism of EAPB0503 in NPM1 mutated AML. It's very well written and methods are described in sufficient details. Some figure quality could be improved (e.g. Fig 3 B, C, D).
Author Response
We thank the reviewer for the positive assessment of our manuscript. We have re-checked the quality of Figure 3 and adjusted the resolution of the scale bars in the confocal microscopy images (Figure 3B). We did not depict any problem of resolution in the remaining panels (images held up in good resolution even at 200%). We hope that the implemented changes meet with your acceptance.

Reviewer 2 Report
Skayneh et al report about some biological consequences of NPM1 mutations in AML and addresses the potential targeting of such mechanisms by using an imidazoquinoxaline derivative.
The methods are well described and the results consistent. The paper is very well written and the results discusse properly. I don't have specific concerns about the manuscript
Author Response
We thank the reviewer for the encouraging remarks.
Reviewer 3 Report
In this paper, the authors confirm the efficacy and selectivity of the imidazoquinoxaline drug EAPB0503 on NPM1mut AML in vivo and in vitro, validating the previously described commitment of p53 pathway and suggesting a mechanism of action involving SENP3, a de-sumoylation protease. This is an interesting study considering the incidence of this genetic aberrancy in AML, which suggests a highly specific mechanism that could lead to a targeted therapeutic opportunity.
From this study it results clear and convincing that EAPB0503 selectively affects NPM1c sumoylation, ubiquitination, and targets SENP3, ARF, HDM2 and p53 axis in NPM1mut AML. However, a better dissection of drug mechanism would strength the relevance of this manuscript and would increase the interest in future potential therapeutic opportunities.
My concerns are:
- About the robustness of the hypothesis:
- Authors sustain that EAP0503 induces the deregulation of SENP3 and the upregulation of ARF. Notably, as shown in figure 4, 6h after treatment SENP3 is not decreased, but ARF is already increased. Thus, considering the complexity of the relation between ARF – SENP3 – NPM1 (ARF requires NPM1 to induce SENP3 degradation, and NPM1 is required for ARF nucleolar localization and stability of both ARF and SENP3), it would be important to unravel this issue, by exploring if drug treatment directly targets SENP3, or differently if affect NPM1c stability, thus restoring NPM1wt allowing a correct folding of ARF that in turn decrease SENP3 level through ubiquitination and proteasomal degradation (Colombo et al., Oncogene 2011). As a matter of fact, SENP3 is related to de-sumoylation, however NPM1c is degraded via proteasome upon ubiquitination, and it is not demonstrated here that NPM1 sumoylation mediates its consequent ubiquitination. Please integrate this part with experiments.
- Line 182: “Treatment with induced SUMOylation of NPM1c, following upregulation of ARF, and downregulation of SENP3, presumably to inhibit ribosomal biogenesis in these cells.” However, drug treatment is demonstrated to restore p53 pathway (Figure 2), whereas no experiments have been done investigating ribosomal biogenesis after treatment. Please sustain this sentence with experimental data or modify the paragraph.
- Considering that the treatment restores p53 pathway, please add in vitro and in vivo experiments testing EAPB0503 combined with chemotherapy at low doses. This would represent interesting evidence in translational context. Moreover, it would strengthen the in vivo survival curve, that in this form (Figure 1B) is not so convincing, considering that the 60% of animals of the treated groups die close to the control group.
- About the quality of the Figures:
- Figure 1B: * of p-value is lacking within the figure
- Western Blot: please improve the quality of HDM2 lanes. In Figure 2B, for example, even if the lane is slight, the edge has a strong signal, indicating a technical problem. Moreover, there is a discrepancy among HDM2 in OCI-AML2 of 2A (slightly expressed) and 2B (relevant signal, even more that in OCI-AML3).
- Figure 2B quantification: P-p53/p53 is hard to interpret, since p53 signal is nearly absent in OCI-AML3. It would be better to split in 2 histograms, that are P-p53/actin and p53/actin.
- Figure 2D-E; Figure 4D-E: it is not clear which patients are NPM1mut and NPM1wt. I think the red ones are the NPM1mut, but it is not clearly stated.
- Figure 3C-D: increase the quality of IP images, since no bands are recognizable.
- The antibody used for NPM1c (PA1 -46356) recognize C-terminal of NPM1mut; considering that OCI-AML3 are described to have a type A mutation, I gather that the antibody recognizes type A mutation. However, it is not reported which NPM1 mutation have the primary samples from patients used in the experiments. Please provide this information, if available.
Author Response
We thank the reviewer for the thorough assessment of our manuscript.
We have answered point by point all his raised comments.
We hope that the implemented changes meet with your acceptance.

Round 2
Reviewer 3 Report
The authors answer and clarify all comments.